# Dynamic Alignment for Multimodal Continual Learning

## Abstract

Multimodal Continual Learning (MMCL) aims to enable models to continuously accumulate knowledge across multiple tasks and modalities without forgetting prior information. MMCL presents more challenges than single-modal continual learning, as it requires effective cooperation and complementarity between modalities. Existing methods often treat modality alignment as a static process, assuming once alignment is established, it remains fixed. However, we argue that modality alignment is inherently dynamic, evolving with task learning and feature propagation across layers. To address this, we introduce Dynamic Alignment Graph Regularization (DAGR), a novel approach that explicitly models the evolving alignment across layers. By incorporating multi-level graph regularization, our method stabilizes the alignment process and mitigates catastrophic forgetting. Extensive experiments on benchmarks, such as MTIL, show that DAGR outperforms static alignment-based methods and other continual learning techniques, achieving superior stability.

## 1 Introduction

Multimodal Continual Learning (MMCL) aims to enable models to continuously accumulate knowledge across multiple tasks and modalities without forgetting previously learned information (Bengio, 2009; Parisi et al., 2019). Compared to single modal continual learning, MMCL presents greater challenges, as it heavily relies on the effective cooperation and complementarity between modalities (Cheng et al., 2020). Traditionally, modality alignment has played a central role in addressing these challenges (Zhang et al., 2021). However, most existing methods focus on alignment that is effective only in the context of multimodal task learning, remaining at a static level. For example, methods such as weight regularization (Kirkpatrick et al., 2017b) and incremental learning strategies that maintain alignment (Rebuffi et al., 2017a) typically enforce alignment at specific layers or representations, neglecting the dynamic adaptability challenges that arise from multi-task learning.

In MMCL, modality alignment is inherently a *dynamic process*, involving both **task-level changes**, such as class and task increments, which are core characteristics of continual learning, and **model-level changes**, such as the propagation of modality information alignment across shallow and deep layers, a key feature of multimodal learning. These two types of changes are interwoven and mutually influential, collectively determining the stability of the alignment process. When the stability of this dynamic process is disrupted, the model may experience instability. For instance, task changes can first induce shifts in the feature distribution of shallow layers, leading to misalignment. As the layers deepen, this , affecting deeper representations, ultimately resulting in erroneous learning guidance during new task learning and exacerbating catastrophic forgetting, as shown in Fig.1. The theoretical derivation of how shallow alignment deviations accumulate and evolve into deep-layer drift can be found in the appendix A.1.

To address this issue, we propose a novel approach called *Dynamic Alignment Graph Regularization (DAGR)*. Unlike traditional methods that treat modality alignment as a rigid constraint, we view it as a dynamic process, focusing on the evolution of fused features across different layers. By explicitly modeling the model-level changes in the alignment process using graph structures, and constraining the internal dynamic changes of the model during multi-task learning, DAGR ensures consistent alignment across layers and tasks, alleviating the accumulation of misalignments in shallow layers and significantly reducing catastrophic forgetting. Specifically, we first construct a Dynamic Align-

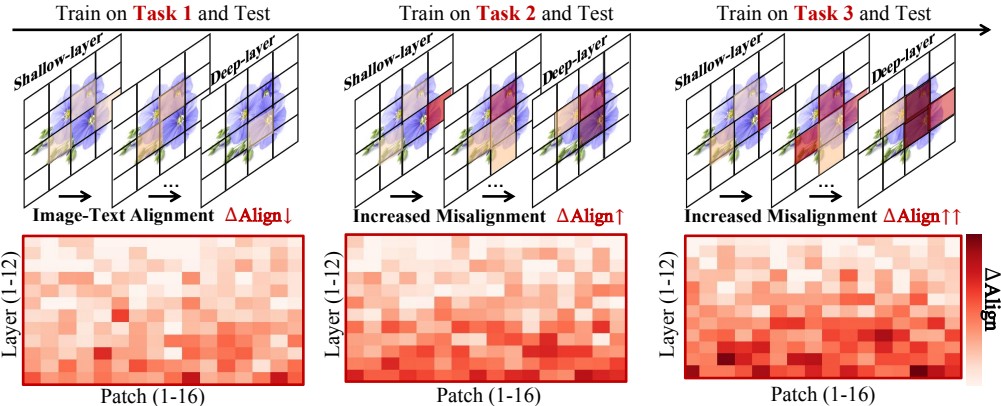

Figure 1: The alignment deviation (ΔAlign) across different layers when the same image is tested after training on different tasks. Shallow misalignments caused by multi-task learning gradually accumulate and affect the deeper layers of the network.

ment Graph for each task, where nodes are generated by clustering the cross-modal features output by the cross-attention modules. Intra-layer edges capture local dependencies, while inter-layer edges model how representations propagate across different layers. We then introduce graph regularization to stabilize the alignment through DAG regularization. This method constrains old subgraphs while allowing new subgraphs to remain flexible. It prevents shallow misalignments from propagating to deeper layers, effectively alleviating catastrophic forgetting and ensuring stable knowledge transfer when learning new tasks.

Our main contributions are summarized as follows:

(1) We propose the concept of *dynamic alignment* in multimodal continual learning, shifting modality alignment from a static constraint to a dynamic, layer-wise evolutionary process that adapts across tasks.

(2) We introduce a novel method called *Dynamic Alignment Graph Regularization (DAGR)*, which captures both intra-layer token interactions and inter-layer representation propagation through a graph-based model. This approach ensures stable and consistent modality alignment across different layers and tasks.

(3) We conduct extensive experiments on benchmark datasets for multimodal continual learning and demonstrate that DAGR significantly improves alignment stability, mitigates catastrophic forgetting, and outperforms existing methods in terms of knowledge retention and adaptability to new tasks.

## 2  RELATED WORK

### 2.1  MULTIMODAL CONTINUAL LEARNING

Existing Multimodal continual learning methods are generally categorized into three types (De Lange et al., 2021). Regularization approaches (Li & Hoiem, 2017; Zhang et al., 2020; Kirkpatrick et al., 2017a; Ahn et al., 2019; Aljundi et al., 2018; Lai et al., 2021) introduce auxiliary loss terms that constrain parameter updates, aiming to preserve knowledge from previous tasks. Architecture-based approaches (Wang et al., 2020; Houlsby et al., 2019) allocate task-specific parameters or expand the network with dedicated components to store task-related knowledge. Rehearsal methods, on the other hand, address forgetting by replaying stored exemplars or utilizing prompt tuning (Jia et al., 2022; Liu et al., 2023; Sohn et al., 2023; Yang et al., 2024). Due to their simplicity and portability, prompt tuning techniques have seen widespread application across various domains (Gao et al., 2023; Ju et al., 2022; Zhou et al., 2022; Khattak et al., 2023; Hegde et al., 2023). However, existing prompt learning methods typically prepend learnable parameters to the original input tokens, which introduces information interference and, ultimately, leads to the loss of pre-trained knowledge during the training process. While these methods show promising results, they primarily focus on mitigating forgetting at the task level and often overlook the dynamic evolution of knowledge and representations across layers during sequential training.

## 2.2 REPRESENTATION DYNAMICS IN MULTIMODAL LEARNING

The dynamics of deep representations have attracted increasing attention in recent years. (Phang et al., 2021) show that fine tuned Transformers form block structured similarity clusters across layers, with deeper layers exhibiting larger changes. (Aitken et al., 2022) demonstrate that adjacent Transformer layers tend to be more similar and that such similarity correlates with prediction confidence. In addition, (Sun et al., 2025) propose LOT Merging to explicitly reduce feature drift by aligning models layer by layer. Collectively, these works indicate that representation dynamics are layer-dependent. Multiple studies visualizing representation trajectories further find that the same input gradually diverges across tasks, and this divergence becomes more pronounced in deeper layers. These findings suggest that errors or alignment biases introduced in early layers may propagate and amplify through the hierarchy like a snowball. Despite these observations, most multimodal learning approaches still treat alignment as a static objective, typically enforced at the top layer or within local attention modules. They ignore the fact that cross modal relations are dynamically established and transmitted through the network hierarchy, rather than being fixed at the output. Building on this insight, we propose a perspective that explicitly models and regularizes how alignment evolves across layers in sequential training scenarios.

## 3 PRELIMINARIES

**Multimodal Continual learning.** MMCL aims to sequentially learn multiple tasks without forgetting previously acquired knowledge. One of the major challenges in CL is catastrophic forgetting, where learning new tasks leads to the loss of knowledge from previously learned tasks. To address this, CL methods attempt to preserve previously learned information while adapting to new tasks. In this paper, we focus on a practical setting called Domain Class Incremental Learning (DCIL), where both the label space and data distribution vary across tasks. Specifically, for a sequence of tasks $\{\mathcal{T}^1, \ldots, \mathcal{T}^N\}$ with corresponding datasets $\{D^i\}$, each task introduces new categories $C^i$ and follows a different distribution $\mathbb{P}(D^i)$, such that $C^i \neq C^j$ and $\mathbb{P}(D^i) \neq \mathbb{P}(D^j)$ for $i \neq j$. In DCIL, the model must learn new categories while retaining the knowledge about previously learned classes and distributions.

**Task-specific prompt learning.** Inspired by recent advances in parameter-efficient tuning (Smith et al., 2023; Hu et al., 2023), a common approach in continual learning is to learn lightweight prompts for each task and store them in a *prompt pool*. These prompts help adapt the frozen model to new tasks without modifying its backbone, making it memory efficient. At inference, the most relevant prompt is retrieved and prepended to the frozen model, guiding it to recall task-specific knowledge and generate better feature embeddings. The prompt selection is done via query-key matching, where at test time, the closest prompt is selected:

$$I_s = \arg\max_{I^i \in \mathbf{I}} \langle f(\boldsymbol{x}), I^i \rangle, \tag{1}$$

where $f(\boldsymbol{x})$ is the input sample's representation.

**Prompt injection in attention modules.** To ensure prompts guide the alignment process, they are injected into each attention module. The prompt is added to the input or intermediate representations, helping the model focus on task-specific aspects while retaining previous knowledge.

**Gaussian distribution modeling.** For each task $T^i$, we model its feature distribution as a Gaussian, with mean $\mu^i$ and covariance $\Sigma^i$. During training, the model learns the distribution parameters:

$$P_{\mathcal{T}^i} = \mathcal{N}(\boldsymbol{x}; \mu^i, \Sigma^i), \tag{2}$$

where $\boldsymbol{x}$ is the feature vector.

**Prompt selection during testing.** During testing, we compute the similarity between the input feature vector $\boldsymbol{x}$ and each task's Gaussian distribution:

$$I_s = \arg\max_{I^i \in \mathbf{I}} \mathcal{D}(f(\boldsymbol{x}), P_{\mathcal{T}^i}), \tag{3}$$

where $\mathcal{D}(\cdot, \cdot)$ is the KL divergence. The closest task distribution is selected, and the corresponding prompt $P_s$ guides the inference process.

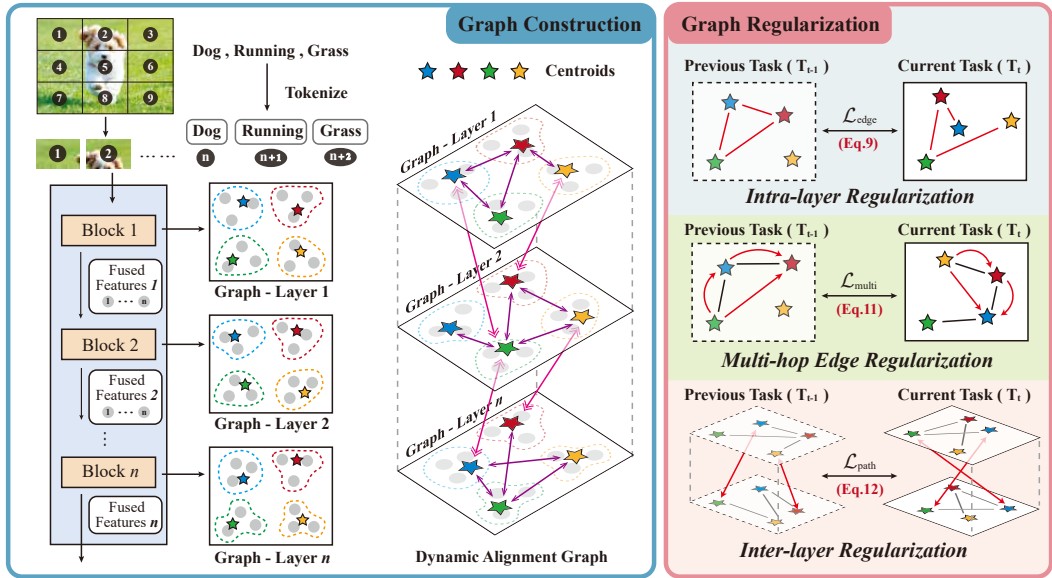

Figure 2: Dynamic Alignment Graph Regularization framework. In Graph Construction, nodes are formed by clustering cross-modal features, with intra-layer edges capturing local dependencies and inter-layer edges modeling feature propagation. In Graph Regularization, three regularizations $L_{\text{edge}}$, $L_{\text{mult}}$, and $L_{\text{path}}$ stabilize alignment by preventing shallow misalignments from propagating to deeper layers, thus mitigating catastrophic forgetting and ensuring stable knowledge transfer.

# 4 METHODOLOGY

A key challenge in multimodal continual learning is capturing how cross-modal alignment evolves across layers during training. To address this, we introduce **Dynamic Alignment Graph Regularization (DAGR)**, as shown in Fig.2, which consists of two main steps. First, we build a **Dynamic Alignment Graph (DAG)** for each task. Nodes are created by clustering cross-modal features from the outputs of cross-attention modules. Intra-layer edges capture local dependencies, while inter-layer edges model how representations propagate across layers. Next, we introduce **Graph Regularization** to stabilize alignment by regularizing the DAG. We constrain old subgraphs while allowing new subgraphs to remain flexible. This approach prevents shallow misalignments from spreading to deeper layers, mitigating catastrophic forgetting and ensuring stable knowledge transfer as new tasks are learned. This prevents shallow misalignments from spreading to deeper layers, mitigating catastrophic forgetting and ensuring stable knowledge transfer as new tasks are learned.

## 4.1 DYNAMIC ALIGNMENT GRAPH

To capture the evolving nature of cross-modal alignment, we construct a DAG $\mathcal{G} = (\mathcal{V}, \mathcal{E})$ for each task. This graph is designed to adapt across various tasks by modeling the interactions between modalities while retaining the knowledge learned in prior tasks. The DAG consists of two essential structures: *intra-layer graphs*, which model local dependencies and interactions within each layer, and *inter-layer graphs*, which capture the propagation of representation information across successive layers. These dual structures work together to enable a comprehensive view of how alignment is continuously updated throughout the network, both within individual layers and across layers.

**Node Construction.** Nodes are derived from the outputs of the cross-attention modules, which naturally combine information from multiple modalities. To reduce the graph's complexity while preserving its semantic richness, token embeddings from each modality are clustered into $K$ centroids per layer. These centroids represent semantically aligned units that help maintain the cross-modal alignment throughout the model. The formulation for node construction is expressed as:

$$\mathcal{V}^{(l)} = \text{Cluster}\Big( \alpha \mathbf{Z}_{\text{img}}^{(l)} + (1 - \alpha)\mathbf{Z}_{\text{txt}}^{(l)}, \ K \Big), \tag{4}$$

where $\mathbf{Z}_{\text{img}}^{(l)}$ and $\mathbf{Z}_{\text{txt}}^{(l)}$ are the feature outputs from the image and text branches at layer $l$ respectively, and $\alpha \in [0, 1]$ is a parameter that controls the fusion ratio between image and text features. Each

centroid $v_k^{(l)} \in \mathcal{V}^{(l)}$ thus serves as a node representing a semantically rich, cross-modally aligned unit that helps preserve the cross-modal interaction across different tasks.

**Intra-layer Edges.** Edges are constructed within a layer to capture token-level dependencies, reflecting the local interactions among tokens. Rather than using simple averaging techniques, we apply a more sophisticated head-wise softmax normalization with a temperature parameter $\beta$. This normalization highlights strong and stable interactions between tokens, ensuring that the local alignment within a layer remains consistent despite task shifts. The mathematical formulation for intra-layer edge construction is given by:

$$e_{ij}^{(l)} = \frac{1}{H} \sum_{h=1}^{H} \frac{\exp\left(\beta \cdot \mathbf{A}_{ij}^{(l,h)}\right)}{\sum_{j'} \exp\left(\beta \cdot \mathbf{A}_{ij'}^{(l,h)}\right)}, \tag{5}$$

where $\mathbf{A}^{(l,h)}$ is the attention matrix for head $h$ at layer $l$, and $H$ represents the number of attention heads. This formulation ensures that token interactions within a layer are captured more accurately, allowing for robust local alignment that is essential for stable task learning.

**Inter-layer Paths.** Edges are constructed to capture how representations propagate from one layer to the next, reflecting the dynamic flow of information across the model. We combine explicit similarity measurements with implicit rollout-based transmission to model this inter-layer flow. The mathematical formulation for inter-layer edge construction is as follows:

$$e_{ij}^{(l \to l+1)} = \gamma \cdot \cos\left(f(v_i^{(l)}), v_j^{(l+1)}\right) + (1 - \gamma) \cdot \mathbf{R}_{ij}^{(l \to l+1)}, \tag{6}$$

where $f(\cdot)$ represents the backbone transformation function, $\cos(\cdot, \cdot)$ denotes cosine similarity, and $\mathbf{R}^{(l \to l+1)}$ is the attention rollout matrix, which models the aggregated influence of a node $v_i^{(l)}$ in layer $l$ on the nodes $v_j^{(l+1)}$ in layer $l+1$. The coefficient $\gamma \in [0, 1]$ is a hyperparameter that controls the balance between direct similarity and indirect rollout-based transmission. Alternatively, inter-layer transitions can be compactly expressed as:

$$\mathbf{T}^{(l \to l+1)} = \text{RowNorm}\left(\mathbf{A}^{(l)} \mathbf{W}^{(l)}\right), \qquad e_{ij}^{(l \to l+1)} = \mathbf{T}^{(l \to l+1)}[i, j], \tag{7}$$

where $\mathbf{W}^{(l)}$ is a projection matrix used to align the representation spaces between adjacent layers. This approach ensures that the flow of information between layers remains stable, further supporting the dynamic alignment process across the network.

## 4.2 GRAPH REGULARIZATION

While the DAG provides a structural foundation for capturing alignment dynamics, it is equally crucial to regularize this process to prevent misalignments from accumulating across tasks, which could lead to catastrophic forgetting. Even small misalignments in shallow layers may propagate and amplify through deeper layers, disrupting the learning process. To address this issue, we introduce a multi-granularity Graph Regularization method that stabilizes the evolution of the DAG while allowing for task-specific adaptability in new classes.

**Subgraph Preparation.** We cluster the cross-attention outputs into prototypes for each task and align them with the prototypes from previous tasks using similarity-based matching. We then construct a binary mask $\mathbf{M}$ to retain only the nodes and edges corresponding to old classes. The binary mask is defined as follows:

$$\mathbf{M}_l[i, j] = \begin{cases} 1, & \text{if edge } (i, j) \text{ belongs to old-class subgraph}, \\ 0, & \text{otherwise}. \end{cases} \tag{8}$$

This ensures that the regularization focuses only on the old-class subgraphs, avoiding excessive rigidity when learning new classes. To improve the stability and interpretability of the graph, we also apply Top-$K$ sparsification to prune weak edges, thereby emphasizing the most critical connections within the network.

**Edge Consistency.** The Edge Consistency regularization ensures that token-to-token interactions remain stable across tasks. Specifically, we penalize the drift in the probability distribution $\mathbf{p}_{l,i}$ associated with each token $i$ in layer $l$ by computing the Kullback-Leibler (KL) divergence between the distributions of consecutive tasks:

$$\mathcal{L}_{\text{edge}} = \sum_l \sum_i \sum_j \mathbf{M}_l[i, j] \, \text{KL}\left(\mathbf{p}_{l,i}^{(t-1)}(j) \parallel \mathbf{p}_{l,i}^{(t)}(j)\right). \tag{9}$$

This term effectively prevents abrupt changes in local interactions, ensuring that token-level dependencies remain stable despite task shifts.

**Multi-hop Edge Consistency.** The Multi-hop Edge Consistency regularization extends the edge consistency constraints to multi-hop transitions. Since errors can accumulate along reasoning chains, it is essential to ensure stability over longer paths. For $k \in \{2, 3\}$, we compute the $k$-step propagation distributions:

$$\mathbf{P}_l^{(t)}(k) = \mathrm{RowNorm}\left(\left(\mathbf{A}_l^{(t)}\right)^k\right), \tag{10}$$

and penalize deviations across tasks by calculating the KL divergence between these multi-hop distributions:

$$\mathcal{L}_{\mathrm{multi}} = \sum_l \sum_i \sum_{k=2,3} \mathrm{KL}\left(\mathbf{P}_l^{(t-1)}(k)[i, \cdot] \parallel \mathbf{P}_l^{(t)}(k)[i, \cdot]\right). \tag{11}$$

This regularization ensures that even indirect interactions remain stable, preventing errors from propagating through the network.

**Path Consistency.** Finally, the Path Consistency regularization enforces consistency in the propagation of information across adjacent layers. By using rollout-based transition matrices $\mathbf{T}_{l \to l+1}^{(t)}$, we constrain the evolution of inter-layer dependencies:

$$\mathcal{L}_{\mathrm{path}} = \sum_l \mathrm{KL}\left(\mathrm{RowNorm}\left(\mathbf{M}_{l \to l+1} \odot \mathbf{T}_{l \to l+1}^{(t-1)}\right) \parallel \mathrm{RowNorm}\left(\mathbf{M}_{l \to l+1} \odot \mathbf{T}_{l \to l+1}^{(t)}\right)\right). \tag{12}$$

This ensures that shallow perturbations do not propagate unchecked, thereby stabilizing the deeper layers of the network.

**Overall Objective.** The overall objective for training is a combination of the Image-Language Matching Loss $\mathcal{L}_{\mathrm{match}}$ and all graph-based regularization terms:

$$\mathcal{L} = \mathcal{L}_{\mathrm{match}} + \lambda_1 \mathcal{L}_{\mathrm{edge}} + \lambda_2 \mathcal{L}_{\mathrm{multi}} + \lambda_3 \mathcal{L}_{\mathrm{path}}, \tag{13}$$

where $\lambda_1, \lambda_2, \lambda_3$ are hyperparameters that control the trade-off between stability and plasticity. By gradually decaying these coefficients across layers (stronger in shallow layers, weaker in deep layers), we achieve further improvements in stability. Since the backbone remains frozen during training, these regularization constraints are realized through the task-specific prompts, which are the only learnable components. This allows for efficient knowledge transfer and prevents catastrophic forgetting, while also enabling the model to learn how to steer the alignment dynamics effectively.

## 5 EXPERIMENTS

**Benchmarks.** To demonstrate the effectiveness of DAGR under the Domain Class incremental learning setting, we conduct experiments on the recently proposed MTIL (Zheng et al., 2023) benchmark. MTIL consists of 11 diverse datasets: Aircraft (Maji et al., 2013), Caltech101 (Fei-Fei et al., 2004), CIFAR100 (Krizhevsky et al., 2009), DTD (Cimpoi et al., 2014), EuroSAT (Helber et al., 2019), Flowers (Nilsback & Zisserman, 2008), Food (Bossard et al., 2014), MNIST (Deng, 2012), OxfordPet (Parkhi et al., 2012), StanfordCars (Krause et al., 2013), and SUN397 (Xiao et al., 2010). This benchmark is highly challenging, involving a total of 1201 classes and significant data distribution shifts across tasks.

**Evaluation metrics.** We adopt two metrics from (Zheng et al., 2023): *Last* and *Avg.*. The *Last* score measures the model performance after completing all tasks, reflecting the final degree of knowledge retention and backward forgetting. The *Avg.* score denotes the average accuracy across all training steps. A lower *Avg.* value usually indicates more severe performance degradation during sequential training, which indirectly reflects the effect of backward forgetting throughout the process. Detailed formulations are provided in the appendixA.2.

**Comparison methods.** We compare DAGR against both full-parameter fine-tuning and parameter-efficient fine-tuning methods. For full fine-tuning, we include ZSCL and ZSCL* (Zheng et al., 2023), LwF (Li & Hoiem, 2017), iCaRL (Rebuffi et al., 2017b), LwF-VR (Ding et al., 2022), and WiSE-FT (Wortsman et al., 2022). For parameter-efficient approaches, we select L2P (Wang et al., 2022c), DualPrompt (Wang et al., 2022b), and S-Prompts (Wang et al., 2022a), DIKI (Tang et al., 2024), which share similar task-specific parameter training procedures with DAGR. Since L2P and DualPrompt were originally developed for ViT (Dosovitskiy et al., 2020), we reproduce them on CLIP and Cross-Attention for fair comparisons.

Table 1: *Avg.*, and *Last* scores (%) of different continue learning methods on MTIL benchmark. † means we reproduce the original methods on vision-language models.

| | Extra data | # Par. | Airc. | Calt. | CIFA. | DTD | Euro. | Flow. | Food | MNIS. | Oxfo. | Cars | SUN. | Average |
|---|---|---|---|---|---|---|---|---|---|---|---|---|---|---|
| Zero-shot | | | 24.8 | 92.9 | 68.4 | 43.8 | 47.7 | 71.4 | 85.8 | 59.5 | 89.1 | 65.8 | 62.6 | 64.7 |
| Upper Bound | | | 62.0 | 96.2 | 89.6 | 79.5 | 98.9 | 97.5 | 92.7 | 99.6 | 94.7 | 89.6 | 81.8 | 89.3 |
| **Avg.** | | | | | | | | | | | | | | |
| LwF | ✓ | 211 M | 36.3 | 86.9 | 72.0 | 59.0 | 73.7 | 60.0 | 73.6 | 74.8 | 80.0 | 37.3 | 58.1 | 64.7 |
| iCaRL | ✓ | 211 M | 35.5 | 89.2 | 72.2 | 60.6 | 68.8 | 70.0 | 78.2 | 62.3 | 81.8 | 41.2 | 62.5 | 65.7 |
| LwF-VR | ✓ | 211 M | 29.6 | 87.7 | 74.4 | 59.5 | 72.4 | 63.6 | 77.0 | 66.7 | 81.2 | 43.7 | 60.7 | 65.1 |
| WiSE-FT | ✓ | 211 M | 26.7 | 86.5 | 64.3 | 57.1 | 65.7 | 58.7 | 71.1 | 70.5 | 75.8 | 36.9 | 54.6 | 60.7 |
| ZSCL* | ✓ | 211 M | 50.7 | 90.9 | 79.8 | 63.8 | 76.6 | 77.3 | 87.0 | 71.9 | 83.0 | 52.0 | 65.9 | 72.6 |
| ZSCL | ✓ | 211 M | 45.1 | 92.0 | 80.1 | 64.3 | 79.5 | 81.6 | **89.6** | 75.2 | 88.9 | 64.7 | 68.0 | 75.4 |
| L2P† | ✗ | 0.5 M | 38.0 | 85.2 | 78.2 | 61.3 | 72.9 | 74.9 | 79.7 | 59.1 | 82.0 | 59.7 | 55.4 | 67.9 |
| DualPmt.† | ✗ | 1.8 M | 37.8 | 84.3 | 78.6 | 60.1 | 71.1 | 73.2 | 79.1 | 73.9 | 82.3 | 55.1 | 52.8 | 68.0 |
| S-Prompts | ✗ | 0.5 M | 37.5 | 92.5 | 77.5 | 58.2 | 76.4 | 74.1 | 78.8 | 57.9 | 83.0 | 60.8 | 54.4 | 68.3 |
| DIKI | ✗ | 1.8 M | 45.1 | 95.5 | 83.1 | 64.8 | 79.9 | 83.5 | 87.0 | 76.2 | 89.6 | 67.0 | 67.1 | 76.3 |
| DAGR | ✗ | 1.8 M | **53.5** | **96.4** | **85.2** | **72.6** | **82.1** | **85.7** | 87.6 | **79.1** | **91.3** | **70.6** | **69.4** | **79.4** |
| **Last** | | | | | | | | | | | | | | |
| LwF | ✓ | 211 M | 26.3 | 87.5 | 71.9 | 66.6 | 79.9 | 66.9 | 83.8 | **99.6** | 92.1 | 66.1 | 80.4 | 74.6 |
| iCaRL | ✓ | 211 M | 35.8 | 93.0 | 77.0 | 70.2 | 83.3 | 88.5 | 90.4 | 86.7 | 93.2 | 81.2 | **81.9** | 80.1 |
| LwF-VR | ✓ | 211 M | 20.5 | 89.8 | 72.3 | 67.6 | 85.5 | 73.8 | 85.7 | **99.6** | 93.1 | 73.3 | 80.9 | 76.6 |
| WiSE-FT | ✓ | 211 M | 27.2 | 90.8 | 68.0 | 68.9 | 86.9 | 74.0 | 87.6 | **99.6** | 92.6 | 77.8 | 81.3 | 77.7 |
| ZSCL* | ✓ | 211 M | 46.0 | 92.3 | 81.2 | 72.4 | 93.0 | 92.1 | 90.8 | **99.6** | 93.3 | **86.6** | 81.7 | 84.5 |
| ZSCL | ✓ | 211 M | 40.6 | 92.2 | 81.3 | 70.5 | 94.8 | 90.5 | **91.9** | 98.7 | 93.9 | 85.3 | 80.2 | 83.6 |
| L2P† | ✗ | 0.5 M | 38.0 | 87.1 | 84.2 | 72.9 | 86.0 | 96.1 | 89.2 | 99.0 | 94.1 | 79.6 | 76.0 | 82.0 |
| DualPmt.† | ✗ | 1.8 M | 37.8 | 87.1 | 84.6 | 71.8 | 89.2 | 96.3 | 89.1 | 99.1 | **94.5** | 79.9 | 76.5 | 82.3 |
| S-Prompts | ✗ | 0.5 M | 37.5 | 95.1 | 83.7 | 70.2 | 97.5 | 96.5 | 89.0 | 99.1 | 94.0 | 79.5 | 75.8 | 83.4 |
| DIKI | ✗ | 1.8 M | 45.2 | 95.7 | 86.3 | 72.9 | 98.0 | 97.0 | 89.2 | 99.4 | 94.2 | 81.6 | 76.6 | 85.1 |
| DAGR | ✗ | 1.8 M | **53.5** | **96.4** | **87.5** | **77.0** | **98.2** | **97.4** | 89.4 | **99.6** | **94.5** | 84.7 | 79.5 | **87.1** |

## 5.1 MAIN RESULTS

Table 1 reports the *Avg.* and *Last* scores of all methods on the MTIL benchmark. "Extra data" indicates the use of memory buffers or reference datasets in methods that rely on distillation (Zheng et al., 2023), and "# Param." denotes the number of trainable parameters. "Upper Bound" is calculated by performing full parameter fine-tuning on each dataset independently, providing a reference point for the achievable *Last* score. As indicated by the bold values, our method consistently achieves the best performance across almost all datasets in both *Avg.* and *Last* scores. In particular, DAGR improves the overall *Avg.* accuracy to 79.4% and the *Last* accuracy to 87.1%, surpassing the strongest baseline DIKI by +3.1% and +2.0% respectively. Compared with existing prompt-based methods such as L2P, DualPrompt, and S-Prompts, DAGR shows clear advantages, demonstrating that simply reusing task-specific prompts is insufficient to handle the severe distribution shifts in MTIL. While ZSCL variants benefit from extra data buffers and achieve competitive performance, our approach outperforms them without relying on any external memory, highlighting the effectiveness of modeling and regularizing information flow through prompts. These results confirm that enforcing graph-level consistency not only stabilizes the continual learning process but also enables prompts to act as effective carriers of task-specific knowledge.

We also conduct experiments on the 16-shot MTIL-FS benchmark (only 16 samples per class). Abbreviated results are shown in Table 2, and the full table can be found in the AppendixA.6. By employing a dynamic alignment graph regularization strategy, along with graph pruning and progressive regularization, our method effectively reduces error accumulation while maintaining strong performance with minimal parameter updates. The results show that, despite using only a small subset of trainable parameters, our method outperforms ZSCL and other full-parameter fine-tuning methods, particularly when training data is limited. Compared to other interruptive prompt tuning methods, our approach exhibits superior stability and competitive performance in data-deficient scenarios.

## 5.2 ANALYSIS

**Ablation.** As shown in Table 3, removing all regularization causes a clear decline in stability and accuracy, while introducing any single loss leads to noticeable improvements. The inter-layer path

Table 2: DAGR on the 16-shot MTIL-FS .

| Method | Avg. | Last | Average |
|--------|------|------|---------|
| ZSCL | 69.3 | 74.0 | 71.7 |
| L2P† | 62.3 | 73.3 | 67.8 |
| DualPrompt† | 64.3 | 74.7 | 69.5 |
| S-Prompts | 63.2 | 73.8 | 68.5 |
| DIKI | 71.9 | 77.1 | 74.5 |
| **DAGR** | **74.2** | **79.3** | **76.8** |

Table 3: Ablation study.

| Method Variant | $L_{edge}$ | $L_{multi}$ | $L_{path}$ | Avg. (%) | Last (%) |
|----------------|------------|-------------|------------|----------|----------|
| w/o Graph Reg. | | | | 71.2 | 79.5 |
| $L_{edge}$ only | ✓ | | | 73.0 | 81.0 |
| $L_{multi}$ only | | ✓ | | 74.1 | 82.3 |
| $L_{path}$ only | | | ✓ | 75.4 | 83.7 |
| $L_{edge} + L_{multi}$ | ✓ | ✓ | | 76.2 | 84.5 |
| $L_{edge} + L_{path}$ | ✓ | | ✓ | 77.0 | 85.2 |
| $L_{multi} + L_{path}$ | | ✓ | ✓ | 77.6 | 85.8 |
| **Full** | ✓ | ✓ | ✓ | **79.4** | **87.1** |

Table 4: *Transfer* scores (%) of different continue learning methods on MTIL benchmark. Metric "transfer" represents the model zero-shot ability retention after being trained on each task.

| | Extra data | # Par. | Airc. | Calt. | CIFA. | DTD | Euro. | Flow. | Food | MNIS. | Oxfo. | Cars | SUN. | Average |
|--|-----------|--------|-------|-------|-------|-----|-------|-------|------|-------|-------|------|------|---------|
| **Transfer** | | | | | | | | | | | | | | |
| LwF | ✓ | 211 M | | 74.5 | 56.9 | 39.1 | **51.1** | 52.6 | 72.8 | 60.6 | 75.1 | 30.3 | 55.9 | 56.9 |
| iCaRL | ✓ | 211 M | | 56.6 | 44.6 | 32.7 | 39.3 | 46.6 | 68.0 | 46.0 | 77.4 | 31.9 | 60.5 | 50.4 |
| LwF-VR | ✓ | 211 M | | 77.1 | 61.0 | 40.5 | 45.3 | 54.4 | 74.6 | 47.9 | 76.7 | 36.3 | 58.6 | 57.2 |
| WiSE-FT | ✓ | 211 M | | 73.5 | 55.6 | 35.6 | 41.5 | 47.0 | 68.3 | 53.9 | 69.3 | 26.8 | 51.9 | 52.3 |
| ZSCL* | ✓ | 211 M | | 78.3 | 64.0 | 42.9 | 45.2 | 63.5 | 84.2 | 56.1 | 78.9 | 44.1 | 64.3 | 62.2 |
| ZSCL | ✓ | 211 M | | 86.0 | 67.4 | **45.4** | 50.4 | **69.1** | 87.6 | 61.8 | 86.8 | 60.1 | **66.8** | 68.1 |
| L2P† | ✗ | 0.5 M | | 65.6 | 50.9 | 30.4 | 41.4 | 49.3 | 71.8 | 36.3 | 77.5 | 55.3 | 53.4 | 53.2 |
| DualPmt.† | ✗ | 1.8 M | | 56.7 | 51.4 | 28.7 | 33.7 | 45.6 | 70.9 | 59.5 | 77.7 | 49.5 | 50.4 | 52.4 |
| S-Prompts | ✗ | 0.5 M | | 67.3 | 49.4 | 26.4 | 39.7 | 47.1 | 70.2 | 34.3 | 78.9 | 56.7 | 52.2 | 52.2 |
| DIKI | ✗ | 1.8 M | | 92.5 | 69.0 | 43.2 | 48.2 | 67.4 | 85.2 | 63.0 | 87.9 | 63.8 | 66.2 | 68.7 |
| DAGR | ✗ | 1.8 M | | **92.9** | **69.2** | 43.9 | 48.7 | 68.2 | 85.6 | **63.6** | **88.9** | **64.7** | 66.5 | **69.3** |

constraint proves to be the most critical, since it directly suppresses the amplification of shallow perturbations as information propagates through the network. When two losses are combined, the gains are larger and more consistent, especially for combinations involving the path constraint, which highlights its central role. The full model that integrates all three losses delivers the best results overall, demonstrating that the different levels of constraints are complementary: one anchors local consistency, another controls propagation within layers, and the third stabilizes cross-layer transitions. Their joint effect produces a more balanced trade-off between stability and plasticity, leading to stronger resistance to forgetting and more reliable continual learning.

**Forward Forgetting.** Although our main experiments focus on backward forgetting and overall performance stability, we also conduct an additional analysis on *forward forgetting*, as shown in Table 4. Forward forgetting measures how well knowledge from earlier tasks transfers to future unseen tasks. We adopt the *Transfer* metric from (Zheng et al., 2023), which evaluates the average accuracy on tasks $i + 1, i + 2, \ldots, N$ after training on task $i$. A lower value indicates stronger forward forgetting. While our method effectively mitigates backward forgetting through task-specific prompts, the absence of task labels during testing and prompt specialization restrict direct adaptation to unseen tasks, detailed analysis in the appendix. To address this, we partially inherit parameters from previous prompts (e.g., initializing task-$i + 1$ prompt with components of task-$i$ prompt), allowing new prompts to retain task-invariant patterns. This inheritance strategy improves forward transfer and enhances zero-shot generalization.

**Heatmap Visualization.** Fig. 3 presents a comparison of attention heatmaps generated by the traditional method and our approach after training on multiple tasks. In the top row, the traditional method exhibits scattered attention, with no clear focus on relevant regions of the image. This diffuse attention is indicative of poor alignment, where the model struggles to retain important features from earlier tasks, leading to catastrophic forgetting. In contrast, the bottom row shows the heatmaps from our method, where attention is concentrated and focused on the key areas. Our approach emphasizes the dynamic evolution of modality alignment, ensuring that important features are consistently retained even as new tasks are learned. By modeling and stabilizing the alignment process across layers, our method effectively mitigates catastrophic forgetting and maintains stable attention, allowing the model to focus on critical regions throughout the learning process.

**Different Task Orders.** Fig. 4 presents a comparison of the performance of the DAGR method and the DualPrompt baseline under different task orders. To assess the impact of task order on model performance, we conducted experiments with sequential learning, reverse learning, and random task order. The results show that the DAGR method maintains relatively stable performance

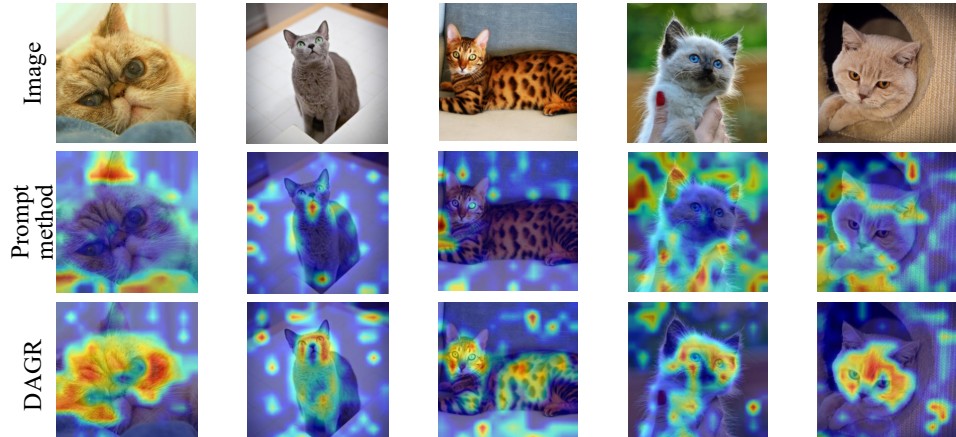

Figure 3: Visualization of forgetting after training on multiple tasks.

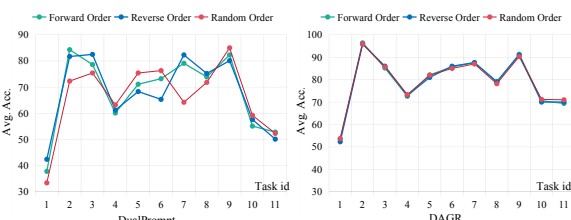

Figure 4: Different task order.

Figure 5: Training costs comparisons. "GPU Mem." denotes the training requirement, and "Acc." is the Avg. scores (%).

| Method | # Param. | Time | GPU Mem. | Acc. |
|---|---|---|---|---|
| ZSCL | 211 M | 12.9 h | 96 GB | 75.4 |
| DIKI | 1.8 M | 2.6 h | 24 GB | 76.3 |
| DAGR | 1.8 M | 2.9 h | 24 GB | 79.4 |
| DAGR(w/GP) | 1.8 M | 2.6 h | 24 GB | 78.8 |
| DAGR(w/PR) | 1.8 M | 2.7 h | 24 GB | 78.6 |
| DAGR(w/Both) | 1.8 M | 2.3 h | 24 GB | 78.0 |

across all task orders, especially in reverse and random task sequences, where the accuracy fluctuations are minimal, demonstrating its robust performance. In contrast, the DualPrompt method exhibits more pronounced performance fluctuations, particularly during task switches, where significant performance degradation is observed. This indicates that the DAGR method, through its dynamic alignment mechanism, effectively mitigates the impact of task order variations, ensuring smoother knowledge transfer between tasks. As a result, DAGR demonstrates superior long-term stability and adaptability compared to DualPrompt. These findings highlight the advantage of DAGR in multi-task sequential learning, where task order uncertainty is present, as it significantly reduces performance fluctuations and ensures more stable learning over time.

**Training Cost Comparison.** To better handle large-scale datasets and prevent excessive regularization from affecting efficiency, we have enhanced our approach with two simple techniques: Graph Pruning (GP) and Progressive Regularization (PR), which improve training efficiency. As shown in Table 5, both techniques slightly decrease accuracy but significantly reduce training time. When applied together in DAGR/(w Both), the accuracy loss is marginal, but the reduction in training time is more pronounced. Despite the small drop in accuracy, performance still surpasses ZSCL and DIKI methods. This demonstrates that a slight accuracy trade-off can lead to a substantial reduction in training time, thus enhancing time efficiency without compromising much performance. These results emphasize the trade-off between accuracy and time efficiency, showing that only a minor accuracy loss is necessary to significantly cut training time costs. Detailed in the AppendixA.4.

## 6 CONCLUSION

We propose a novel DAGR method to tackle catastrophic forgetting in multimodal continual learning. Unlike traditional static alignment methods, DAGR models the dynamic evolution of cross-layer alignment and applies multi-level graph regularization to stabilize the process. Experimental results show DAGR outperforms existing methods, particularly in stability and mitigating catastrophic forgetting. However, challenges remain in handling large-scale datasets and efficiently managing multiple tasks. While graph pruning and progressive regularization help, further optimization is needed. Future work could explore integrating DAGR with open-set learning to improve performance in complex task environments.

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

# A   APPENDIX

## A.1   THEORETICAL ANALYSIS

We provide theoretical insights into how shallow misalignment can accumulate into severe deep-layer drift, and how our proposed graph regularization constrains such dynamics. We formalize the alignment process as a sequence of stochastic transition operators and analyze their sensitivity under perturbations.

**Definition 1** (Layer-wise Transition Matrices). For each layer $l$, denote the intra-layer transition operator (row-stochastic) by $\Pi_l \in \mathbb{R}^{K_l \times K_l}$, constructed from normalized attention weights. The inter-layer transition from layer $l$ to $l+1$ is denoted by $\Upsilon_{l \to l+1} \in \mathbb{R}^{K_l \times K_{l+1}}$.

**Definition 2** (Composite Transition Operator). The effective propagation from the input layer to layer $L$ is modeled as

$$\mathcal{T}_{1 \to L} = \Pi_1^{k_1} \, \Upsilon_{1 \to 2} \, \Pi_2^{k_2} \, \Upsilon_{2 \to 3} \cdots \Pi_{L-1}^{k_{L-1}}. \tag{14}$$

Here $k_l$ denotes the number of intra-layer propagation steps considered for layer $l$.

**Lemma 3** (Perturbation Notation). *Let $\Pi_l^*$ and $\Upsilon_{l \to l+1}^*$ denote the reference (previous-task) operators. Define perturbations:*

$$\Delta \Pi_l = \Pi_l - \Pi_l^*, \qquad \Delta \Upsilon_{l \to l+1} = \Upsilon_{l \to l+1} - \Upsilon_{l \to l+1}^*.$$

**Theorem 4** (Single-Layer multi Hop Sensitivity). *For any layer $l$ and propagation depth $k$, we have*

$$\|\Pi_l^k - (\Pi_l^*)^k\| \;\le\; k \cdot \|\Delta \Pi_l\| \cdot \max\{\|\Pi_l\|, \|\Pi_l^*\|\}^{k-1}, \tag{15}$$

*where $\|\cdot\|$ is any submultiplicative matrix norm (e.g., $\|\cdot\|_1$).*

*Proof.* Expand $(\Pi_l^* + \Delta \Pi_l)^k - (\Pi_l^*)^k$ using the telescoping series:

$$(A + B)^k - A^k = \sum_{i=0}^{k-1} A^{k-1-i} B (A + B)^i.$$

Apply submultiplicativity of the matrix norm and bound by $k \|B\| \max\{\|A\|, \|A + B\|\}^{k-1}$. $\square$

**Corollary 5** (Accumulation in Deeper Layers). *Even if $\|\Delta \Pi_l\|$ is small, the deviation in $\Pi_l^k$ grows linearly in $k$. Thus, shallow misalignment may be small at one step but becomes significant after multi hop propagation.*

**Theorem 6** (Cross-Layer Perturbation Expansion). *For the composite operator $\mathcal{T}_{1 \to L}$,*

$$\|\mathcal{T}_{1 \to L} - \mathcal{T}_{1 \to L}^*\| \;\le\; \sum_{b=1}^{B} \left( \prod_{j<b} \|M_j^*\| \right) \|\Delta M_b\| \left( \prod_{j>b} \|M_j\| \right), \tag{16}$$

*where $\{M_j\}_{j=1}^B$ are the intra- and inter-layer factors in $\mathcal{T}_{1 \to L}$.*

*Proof.* Apply a first-order perturbation expansion to the product $\prod_{j=1}^B M_j$, writing $M_j = M_j^* + \Delta M_j$. Each perturbation contributes an additive term scaled by the product of norms of all other factors. $\square$

**Corollary 7** (Shallow-to-Deep Amplification). *Perturbations in shallow operators ($\Delta M_b$ with small $b$) are amplified by the product of all subsequent layer norms. This explains why shallow misalignment accumulates and manifests more severely at deeper layers.*

**Theorem 8** (Pinsker Bound for Edge Regularization). *For each row distribution $\Pi_l(i, \cdot)$ and reference $\Pi_l^*(i, \cdot)$,*

$$\|\Pi_l(i, \cdot) - \Pi_l^*(i, \cdot)\|_1 \;\le\; \sqrt{2 \, \mathrm{KL}(\Pi_l^*(i, \cdot) \,\|\, \Pi_l(i, \cdot))}. \tag{17}$$

**Corollary 9** (Effect of Edge KL Loss). *Minimizing $\mathcal{L}_{edge} = \sum_i \mathrm{KL}(\Pi_l^*(i, \cdot) \,\|\, \Pi_l(i, \cdot))$ ensures bounded $L_1$ deviation row-wise, which in turn reduces the upper bound in Theorem 1.*

**Theorem 10** (multi Hop Regularization). *Let* $\mathbf{P}_l^{(t)}(k) = RowNorm((\Pi_l^{(t)})^k)$. *Then,*

$$\|\mathbf{P}_l^{(t)}(k) - \mathbf{P}_l^{(t-1)}(k)\|_1 \leq \sqrt{2\,\mathcal{L}_{path}}, \tag{18}$$

*where* $\mathcal{L}_{path}$ *is the KL divergence between* $k$*-hop distributions.*

**Corollary 11** (Error Containment by Path Loss). *Even if edge-level constraints allow small drift, path-level regularization directly prevents multi hop amplification, thereby containing error chains.*

**Theorem 12** (Inter-Layer Regularization). *For rollout-based transition matrices* $\mathbf{T}_{l\to l+1}^{(t)}$,

$$\|\mathbf{T}_{l\to l+1}^{(t)} - \mathbf{T}_{l\to l+1}^{(t-1)}\|_1 \leq \sqrt{2\,\mathcal{L}_{inter}}. \tag{19}$$

*Remark* 13. This constraint stabilizes the transmission of information across layers, preventing mid-layer disruptions from being magnified downstream.

**Proposition 14** (Effect of Top-$K$ Sparsification). *Applying Top-$K$ sparsification to each row of* $\Pi_l$ *projects distributions onto a* $K$*-sparse simplex. This reduces the maximum attainable* $\|\Delta\Pi_l\|$, *thereby tightening the perturbation bounds in Theorems 1 and 2.*

**Theorem 15** (Overall Stability). *Combining the above results, the deviation between* $\mathcal{T}_{1\to L}$ *and* $\mathcal{T}_{1\to L}^*$ *is bounded by a weighted sum of controlled terms:*

$$\|\mathcal{T}_{1\to L} - \mathcal{T}_{1\to L}^*\| \leq g(\mathcal{L}_{edge}, \mathcal{L}_{path}, \mathcal{L}_{inter}),$$

*where* $g(\cdot)$ *is monotone increasing in each argument. Thus, minimizing the proposed graph regularizers provably reduces alignment drift and error accumulation.*

## A.2 METRICS

Here we formulate the *Transfer*, *Avg.* and *Last* metrics.

Assume that $p_j^{(i)}$ is the model's accuracy on task $j$ after being trained on task $i$, then the Transfer, Avg and Last metrics for task $j$ can be calculated as:

$$
\begin{aligned}
\text{Transfer}_j &= \frac{1}{j-1}\sum_{i=1}^{j-1} p_j^{(i)}, \quad j = 2, 3, \cdots, N \\
\text{Avg}_j &= \frac{1}{N}\sum_{i=1}^{N} p_j^{(i)}, \quad j = 1, 2, \cdots, N \\
\text{Last}_j &= p_j^{(N)}, \quad j = 1, 2, \cdots, N
\end{aligned}
\tag{20}
$$

where $N$ is the number of tasks. It's clear that *Transfer* metric can indicate the zero-shot capability while *Last* metric shows the extent of backward forgetting.

## A.3 FORWARD FORGETTING ANALYSIS

**Prompt inheritance.** Let $P^{(i)} \in \mathbb{R}^{m\times d}$ be the prompt parameters (e.g., $m$ tokens, width $d$) after finishing task $i$. Before training task $i+1$, we *partially inherit* $P^{(i)}$ to initialize $P^{(i+1)}$:

$$P_0^{(i+1)} = M \odot \left(\rho\, P^{(i)} + (1-\rho)\, \tilde{P}^{(i+1)}\right) + (\mathbf{1}-M) \odot \tilde{P}^{(i+1)}, \tag{21}$$

where $M \in \{0,1\}^{m\times d}$ is a binary mask selecting the "shared" (task-invariant) components to inherit, $\rho \in [0,1]$ controls the inheritance strength, $\tilde{P}^{(i+1)}$ is a standard initializer (e.g., Xavier/normal), and $\odot$ denotes the Hadamard product. Equation equation 21 copies a convex combination of old parameters into the masked (shared) part while freshly initializing the task-specific remainder.

**Training objective for task** $i+1$. During task $i+1$, only prompts are trainable while the backbone remains frozen. Starting from $P_0^{(i+1)}$, we optimize

$$\mathcal{L}^{(i+1)} = \mathcal{L}_{\text{task}}^{(i+1)} + \lambda_{\text{inh}} \underbrace{\left\|M \odot \left(P^{(i+1)} - P^{(i)}\right)\right\|_2^2}_{\mathcal{L}_{\text{inherit}}} + \lambda_1 \mathcal{L}_{\text{edge}} + \lambda_2 \mathcal{L}_{\text{multi}} + \lambda_3 \mathcal{L}_{\text{path}}, \tag{22}$$

where $\mathcal{L}_{\text{task}}^{(i+1)}$ is the supervised objective on $T_{i+1}$, $\mathcal{L}_{\text{inherit}}$ softly anchors the inherited subset to retain task-invariant patterns, and $\mathcal{L}_{\text{edge}}, \mathcal{L}_{\text{multi}}, \mathcal{L}_{\text{path}}$ are the graph regularizers used in the main method.

(i) The initialization in equation 21 encourages zero-shot retention and better forward transfer by seeding the new prompt with stable features while keeping sufficient plasticity in the unmasked part. (ii) The inheritance penalty in equation 22 can be annealed over epochs or layers to balance stability/plasticity. (iii) When $M = \mathbf{1}$ the scheme reduces to a global convex inheritance $P_0^{(i+1)} = \rho P^{(i)} + (1 - \rho)\tilde{P}^{(i+1)}$; when $M = \mathbf{0}$ it becomes standard fresh initialization.

## A.4 TRAINING COST COMPARISON

**Baseline cost.** Let $T$ denote the training time, $M$ the GPU memory requirement, and $A$ the achieved average accuracy. For a model with trainable parameter size $p$, the baseline cost can be abstracted as

$$T \propto \sum_{l=1}^{L} \left( C_{\text{attn}}^{(l)} + C_{\text{reg}}^{(l)} \right), \quad M \propto \sum_{l=1}^{L} S^{(l)}, \tag{23}$$

where $C_{\text{attn}}^{(l)}$ is the complexity of attention in layer $l$, $C_{\text{reg}}^{(l)}$ is the cost introduced by regularization (e.g., graph-based constraints), and $S^{(l)}$ is the memory footprint of layer $l$. In ZSCL (full fine-tuning), $p \approx 211M$, which dominates both $T$ and $M$. In DAGR and DIKI, $p \approx 1.8M$ (prompt parameters only), leading to significantly lower costs.

**Graph pruning.** In DAGR, each task produces a Dynamic Alignment Graph (DAG) with $K$ nodes per layer, leading to $O(K^2)$ intra-layer edges and $O(K^2)$ inter-layer edges. To reduce cost, we apply Top-$k$ sparsification:

$$A_{i,j}^{(l)} \leftarrow \begin{cases} A_{i,j}^{(l)}, & j \in \text{Top-}k\left(\{A_{i,j'}^{(l)}\}_{j'}\right), \\ 0, & \text{otherwise}, \end{cases} \tag{24}$$

which reduces the number of active edges from $O(K^2)$ to $O(K \cdot k)$. The theoretical time reduction:

$$T_{\text{prune}} \approx \frac{k}{K} T_{\text{full}}, \tag{25}$$

while memory footprint is reduced proportionally by zeroing weak connections.

**Progressive regularization.** Instead of applying all regularization terms with fixed weights $(\lambda_1, \lambda_2, \lambda_3)$ from the beginning, we gradually increase their influence across epochs $e = 1, \ldots, E$:

$$\lambda_i(e) = \lambda_i^{\max} \cdot \frac{e}{E}, \quad i \in \{1, 2, 3\}. \tag{26}$$

This strategy reduces the regularization overhead in early epochs, thus shortening convergence time:

$$T_{\text{prog}} \approx T_{\text{task}} + \frac{1}{E} \sum_{e=1}^{E} \sum_{i=1}^{3} \lambda_i(e) C_i, \tag{27}$$

where $C_i$ is the cost coefficient of each regularizer. Empirically, progressive weighting allows faster warm-up and lowers effective training time without severely impacting final accuracy.

**Overall efficiency.** Combining graph pruning and progressive regularization yields

$$T_{\text{DAGR/w(A,B)}} \approx \frac{k}{K} T_{\text{full}} + T_{\text{task}} + \frac{1}{E} \sum_{e=1}^{E} \sum_{i=1}^{3} \lambda_i(e) C_i, \tag{28}$$

which explains why DAGR/w(A,B) reduces training time from 2.9h to 2.3h with only a marginal drop in accuracy, as observed in Table 5. The trade-off is controlled by $k/K$ and the annealing schedule of $\lambda_i(e)$.

## A.5   K Ablation Study

To assess the impact of the clustering number $K$ on model performance, we conducted an ablation study by adjusting the clustering number $K$ used in graph regularization and observing its effect on the model's performance. Specifically, we performed experiments on multiple tasks using different $K$ values (ranging from 3 to 7) to generate graph structures, and recorded the model's performance in terms of Avg.(%) and Last(%).

The clustering number $K$ influences the number of nodes in the graph structure at each layer. Smaller values of $K$ may result in overly coarse nodes that fail to capture fine-grained task-specific details, while larger values of $K$ may increase computational costs and introduce more noise. Through this experiment, we aim to find an optimal balance that ensures graph regularization effectively stabilizes alignment relationships and mitigates catastrophic forgetting during multi-task learning. The results of the experiment are shown in the table below:

| Clustering Number $K$ | Avg. (%) | Last. (%) |
|---|---|---|
| 3 | 74.2 | 77.4 |
| 4 | 75.1 | 78.1 |
| 5 | 76.5 | 78.6 |
| 6 | 77.3 | 79.4 |
| 7 | 76.8 | 79.0 |

Table 5: Performance with Different Clustering Numbers $K$.

Thus, the results suggest that $K = 6$ is a reasonable choice, providing a good balance between computational efficiency and optimal alignment stability and performance. Through this ablation study, we found that the clustering number $K$ significantly impacts the model's performance. An appropriate clustering number improves the alignment stability and model performance.

### Analysis of Task Label Presence and Model zero-shot Adaptability

In multi-task learning, the presence or absence of task labels during testing plays a crucial role in the model's ability to adapt to unseen tasks. The adaptation process is especially critical when dealing with task-specific prompts, where the model generates task-specific representations based on the prompts it receives. When there is no task label during testing, the model faces difficulties in selecting the appropriate prompt (Algorithmic constraints), which can significantly impact its performance on new, unseen tasks. This section analyzes the effects of task labels on model adaptability and the solution introduced by our Inheritance Strategy.

**Without Task Labels: The Specialization Problem** In our experiments, the model's specialization is driven by the use of task-specific prompts. These prompts are designed to tailor the model's attention and decision-making process to each specific task. However, when testing without task labels, the model may face situations where the task is assigned an incorrect prompt. This error in prompt assignment can severely limit the model's ability to generalize to the new task, as it will try to apply a specialized prompt designed for another task. Specifically, the issues that arise when task labels are absent include:

- **Over-Specialization:** Task-specific prompts are highly specialized for individual tasks, and without task labels to guide prompt assignment, the model may use an incorrect prompt. This incorrect assignment leads to poor task adaptation because the specialized prompt doesn't align with the new task's requirements.

- **Forward Forgetting:** The lack of task-specific information can exacerbate the problem of forward forgetting, where knowledge from earlier tasks fails to transfer effectively to future unseen tasks. The model may struggle to retain task-invariant knowledge because it is locked into a specialized prompt designed for a specific task.

**The Role of Inheritance Strategy** To address the challenges posed by the absence of task labels and the specialization problem, we introduce an Inheritance Strategy. This strategy partially inherits parameters from previous prompts when learning new tasks, thereby retaining task-invariant patterns. By initializing the prompt for task $i + 1$ with components of task $i$'s prompt, the model can leverage

prior knowledge and maintain a degree of consistency across tasks. The benefits of the Inheritance Strategy include:

- **Task-Invariant Knowledge Sharing:** Instead of completely overwriting the prompt for the new task, the model inherits components from the previous task's prompt, ensuring that shared patterns or task-invariant knowledge are preserved. This helps the model generalize better to new tasks, even without explicit task labels.
- **Improved Forward Transfer:** The inheritance of parameters ensures that new prompts can retain relevant knowledge from previous tasks, improving **forward transfer**—the ability of the model to transfer knowledge from prior tasks to future tasks. This also reduces the impact of forward forgetting.
- **Zero-Shot Generalization:** By maintaining task-invariant knowledge through prompt inheritance, the model is better equipped for zero-shot generalization. This means the model performs well on unseen tasks even when task labels are not available, as it can infer common patterns from previous tasks.

**With Task Labels: No Need for Inheritance Strategy** When task labels are available during testing, the model can easily distinguish between different tasks and select the correct prompt for each task. Thus, task-specific prompts can be directly assigned based on the provided task label, which alleviates the problem of prompt misassignment. The key points when task labels are present are:

- **No Specialization Problem:** With task labels, the model is directly informed of the task at hand, allowing it to select the appropriate prompt without error. This eliminates the issue of over-specialization caused by incorrect prompt assignment.
- **Reduced Need for Inheritance:** Since the model can directly use the correct prompt based on the task label, there is no need for the additional algorithmic complexity introduced by the Inheritance Strategy. The model can adapt directly to new tasks without the need for parameter inheritance.
- **Easier Adaptation:** The availability of task labels makes it easier for the model to adapt to new tasks, as it can use the full potential of task-specific prompts without worrying about prompt misassignment or task incompatibility.

In summary, the absence of task labels during testing creates significant challenges for task adaptation, particularly due to the specialization of prompts. In these cases, the Inheritance Strategy helps by partially inheriting parameters from previous tasks, allowing the model to retain task-invariant patterns and improve forward transfer. However, when task labels are available, the model can directly assign the correct prompts to each task, and the additional complexity of the Inheritance Strategy becomes unnecessary. This highlights the importance of task labels in facilitating task adaptation and reducing the need for specialized algorithms when task identification is clear. **Key Takeaway:** The Inheritance Strategy is essential when task labels are not available, but when task labels are provided, the model can efficiently adapt to new tasks without the need for such a strategy.

## A.6    6-SHOT MTIL-FS BENCHMARK

Table.6 provides full results of different continue learning methods on our modified 16-shot MTIL-FS benchmark. Our DAGR shows consistent improvements compared to previous methods.

## A.7    ALGORITHMIC DETAILS

In this appendix, we provide the complete training and inference procedure of our framework (Algorithm 1). During training, the backbone network is kept frozen, and only the corresponding prompt parameters are optimized for each new task. By utilizing clustering and attention mechanisms, we construct graph structures at each layer, and the optimization of the prompts is regularized by layer-internal edges, multi-hop consistency, and inter-layer paths. This allows for the stabilization of knowledge across layers and progressive adaptation to new tasks. During inference, the model selects the most suitable prompt based on the feature statistics and Gaussian distributions computed from the training phase, and injects it into the frozen backbone network for prediction. This design ensures both efficient inference and compatibility across tasks, effectively alleviating forgetting while maintaining scalability in various continual learning settings.

Table 6: Full results of different continue learning methods on 16-shot MTIL-FS benchmark. †means we reproduce the original methods on vision-language models.

| | Extra data | #Param. | Aircraft | Caltech101 | CIFAR100 | DTD | Flowers | Food | Cars | SUN397 | Average |
|---|---|---|---|---|---|---|---|---|---|---|---|
| Zero-shot | | | 24.8 | 92.9 | 68.4 | 43.8 | 71.4 | 85.8 | 65.8 | 62.6 | 64.4 |
| Upper Bound | | | 62.0 | 96.2 | 89.6 | 79.5 | 97.5 | 92.7 | 89.6 | 81.8 | 86.1 |
| **Avg.** | | | | | | | | | | | |
| ZSCL | ✓ | 211 M | 33.5 | 90.5 | 74.7 | 58.5 | 79.7 | 87.7 | 64.8 | 64.8 | 69.3 |
| L2P† | × | 0.5 M | 30.2 | 84.5 | 70.1 | 51.9 | 69.6 | 77.1 | 60.0 | 55.2 | 62.3 |
| DualPmt.† | × | 1.8 M | 36.5 | 89.5 | 72.5 | 52.7 | 72.3 | 80.8 | 56.1 | 54.2 | 64.3 |
| S-Prompts | × | 0.5 M | 30.6 | 86.8 | 70.0 | 51.7 | 74.3 | 78.5 | 60.7 | 53.0 | 63.2 |
| DIKI | × | 1.8 M | f41.3 | 95.3 | 76.5 | 58.5 | 82.2 | 86.4 | 68.2 | 66.6 | 71.9 |
| DAGR | × | 1.8 M | **49.3** | **95.6** | **78.5** | **62.7** | **85.4** | **88.1** | **68.4** | **69.1** | **74.6** |
| **Last** | | | | | | | | | | | |
| ZSCL | ✓ | 211 M | 27.7 | 90.9 | 74.4 | 64.7 | 90.2 | **89.2** | **80.6** | 74.6 | 74.0 |
| L2P† | × | 0.5 M | 30.2 | 87.1 | 75.4 | 64.7 | 91.9 | 86.4 | 76.1 | 74.7 | 73.3 |
| DualPmt.† | × | 1.8 M | 36.5 | 91.0 | 75.1 | 65.1 | 92.9 | 86.2 | 76.2 | 74.2 | 74.7 |
| S-Prompts | × | 0.5 M | 30.6 | 89.2 | 75.8 | 63.8 | 93.9 | 86.2 | 76.7 | 73.9 | 73.8 |
| DIKI | × | 1.8 M | 41.3 | 95.6 | 79.0 | 67.3 | 94.4 | 86.8 | 77.6 | 74.4 | 77.1 |
| DAGR | × | 1.8 M | **49.0** | **95.8** | **81.2** | **69.5** | **94.6** | 88.0 | 79.8 | **76.1** | **79.3** |

## A.8 VISUALIZATION OF CROSS-TASK ALIGNMENT DYNAMICS

Figure 6 illustrates the evolution of graph alignment when testing Task 1 after sequentially learning new tasks. Each panel corresponds to a different stage: (i) after training on Task 1 and testing on Task 1 (T1→Test1), (ii) after training on Task 2 and testing on Task 1 (T2→Test1), and (iii) after training on Task 3 and testing on Task 1 (T3→Test1).

In each graph, nodes denote clustered token representations, and edges represent inter-node relationships derived from attention distributions. The edge thickness and color intensity indicate the strength of alignment, while the node color corresponds to the magnitude of $\Delta$Align, reflecting how much the representation of each node changes relative to the previous stage. As more tasks are introduced, we observe a gradual shift in both node activations and inter-layer connectivity: several nodes become darker and edges more pronounced, highlighting regions where knowledge of Task 1 is altered due to subsequent task learning.

This visualization confirms that continual learning leads to non-trivial redistribution of attention and feature alignment, which explains potential forgetting effects. It also validates the necessity of the proposed regularization, as it constrains these shifts to stabilize cross-task relationships.

## A.9 WRITING PROCESS ENHANCEMENT WITH LARGE MODELS

In compliance with ICLR guidelines, we disclose that large models were incorporated into our writing process to improve grammar and syntax. The goal was to produce a fluent and well-structured manuscript that adheres to academic writing conventions, ultimately enhancing the readability and impact of our work.

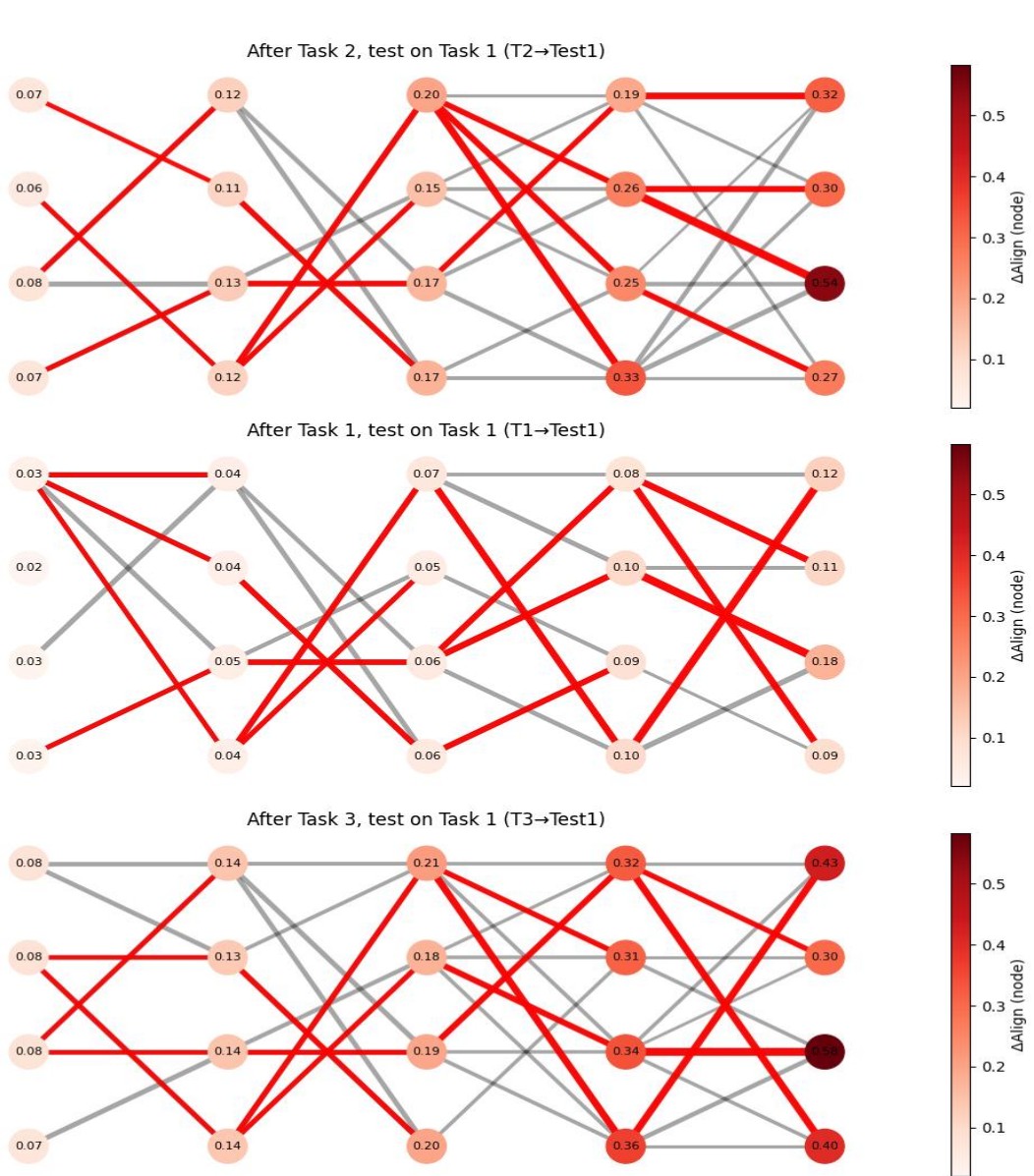

Figure 6: Cross-Task Alignment Dynamics. This figure shows the evolution of alignment in Task 1 after learning Tasks 1, 2, and 3. Nodes represent token clusters, with edge thickness and color indicating alignment strength, and node color reflecting ΔAlign changes. The results highlight the redistribution of attention and potential forgetting, emphasizing the need for regularization.

---

**Algorithm 1** DAGR: Dynamic Alignment Graph Regularization for Multimodal Continual Learning

---

**Require:** Frozen backbone $\mathcal{F}$ with $L$ cross-attention layers; task stream $\{\mathcal{T}_1, \ldots, \mathcal{T}_N\}$; prompt pool $\mathcal{P} = \{\}$; cluster number $K$; fusion ratio $\alpha$; temperature $\beta$; inter-layer balance $\gamma$; regularization weights $(\lambda_1, \lambda_2, \lambda_3)$; multi-hop set $\mathcal{K} = \{2, 3\}$; optional *Top-K sparsification* and *progressive regularization*.

**Ensure:** Trained prompt pool $\mathcal{P}$; task-wise Gaussian stats $\{(\mu_i, \Sigma_i)\}_{i=1}^N$.

1: **function** TRAIN-DAGR($\{\mathcal{T}_i\}_{i=1}^N$)
2:     **for** $i = 1$ **to** $N$ **do**                                             $\triangleright$ Sequential tasks (DCIL)
3:         Initialize prompt $P_i$ (e.g., inherit partially from $P_{i-1}$ if $i > 1$).
4:         $\mu_i, \Sigma_i \leftarrow$ estimate Gaussian of task-$i$ features (running mean/cov on minibatches).
5:         **for each** minibatch $(x^{img}, x^{txt}, y) \sim \mathcal{T}_i$ **do**
6:             Inject $P_i$ into every cross-attention layer of $\mathcal{F}$ and forward pass to get per-layer fused tokens:
7:                 $Z_{img}^{(l)}, Z_{txt}^{(l)}$ for $l = 1 \ldots L$.
8:             **Node construction (per layer):**
9:                 $V^{(l)} \leftarrow \mathrm{Cluster}\big(\alpha Z_{img}^{(l)} + (1 - \alpha) Z_{txt}^{(l)},\ K\big)$
10:             **Intra-layer edges (per layer):**
11:                 $A^{(l,h)} \leftarrow$ attention matrix of head $h$; $e_{ij}^{(l)} \leftarrow \frac{1}{H} \sum_h \mathrm{softmax}_j(\beta A_{ij}^{(l,h)})$
12:             **Inter-layer transitions (between $l \to l+1$):**
13:                 $R^{(l \to l+1)} \leftarrow$ attention rollout; $e_{ij}^{(l \to l+1)} \leftarrow \gamma \cdot \cos(f(v_i^{(l)}), v_j^{(l+1)}) + (1 - \gamma) \cdot R_{ij}^{(l \to l+1)}$
14:             (Optional) Row-normalize and apply Top-$K$ sparsification to $E^{(l)}$ and $T^{(l \to l+1)}$.
15:             **Old-class masking**: build $M^{(l)}$ and $M^{(l \to l+1)}$ to keep only edges/paths of old classes.
16:             **Losses:**
17:                 $\mathcal{L}_{\text{task}} \leftarrow$ CE on logits (or task head) from $\mathcal{F}$ with $P_i$.
18:                 $\mathcal{L}_{\text{edge}} \leftarrow \sum_l \sum_i \sum_j M_{ij}^{(l)} \mathrm{KL}\big(p_{l,i}^{(t-1)}(j) \,\|\, p_{l,i}^{(t)}(j)\big)$
19:                 $\mathcal{L}_{\text{multi}} \leftarrow \sum_l \sum_i \sum_{k \in \mathcal{K}} \mathrm{KL}\big(P_l^{(t-1)}(k)[i, \cdot] \,\|\, P_l^{(t)}(k)[i, \cdot]\big)$
20:                 $\mathcal{L}_{\text{path}} \leftarrow \sum_l \mathrm{KL}\big(\mathrm{RowNorm}(M^{(l \to l+1)} \odot T_{l \to l+1}^{(t-1)}) \,\|\, \mathrm{RowNorm}(M^{(l \to l+1)} \odot T_{l \to l+1}^{(t)})\big)$
21:                 **Total**: $\mathcal{L} \leftarrow \mathcal{L}_{\text{task}} + \lambda_1 \mathcal{L}_{\text{edge}} + \lambda_2 \mathcal{L}_{\text{multi}} + \lambda_3 \mathcal{L}_{\text{path}}$
22:                 (Optional) Progressive regularization: decay $(\lambda_1, \lambda_2, \lambda_3)$ w.r.t. layer depth.
23:                 Update only $P_i$ (backbone $\mathcal{F}$ is frozen) by $\nabla_{P_i} \mathcal{L}$.
24:         **end for**
25:         $\mathcal{P} \leftarrow \mathcal{P} \cup \{P_i\}$ **and** store $(\mu_i, \Sigma_i)$ for prompt selection.
26:     **end for**
27: **end function**

28: **function** INFER-DAGR($x^{img}, x^{txt}$)
29:     Extract feature $f(x)$; select prompt index
30:         $s \leftarrow \arg\max_{i \in \{1..N\}} D\big(f(x), \mathcal{N}(\mu_i, \Sigma_i)\big)$                    $\triangleright$ e.g., KL/Euc. distance
31:     Inject $P_s$ into $\mathcal{F}$; forward to obtain logits $\hat{y}$ and prediction.
32:     **return** $\hat{y}$
33: **end function**