# OpenReview forum: "Dynamic Alignment for Multimodal Continual Learning"
_ICLR.cc/2026/Conference — ICLR 2026 Conference Withdrawn Submission_

### Official Review · Reviewer_vw7M · 2025-10-29

**Soundness:** 3
**Presentation:** 3
**Contribution:** 3
**Rating:** 4
**Confidence:** 3

**Summary:**

The paper develops a method for addressing catastrophic forgetting in multimodal continual learning (MMCL) setting. The motivation is that prior methods treat modality alignment as a static process, assuming once alignment is achieved, it remains fixed. In contrast, the paper shows that alignment is dynamic, evolving across layers and tasks as representations change over time. To address this process, the paper proposes Dynamic Alignment Graph Regularization (DAGR) to model how cross-modal features interact through the network hierarchy. DAGR constructs a Dynamic Alignment Graph (DAG) for each task. In this graph, the nodes represent clustered cross-modal feature centroids from cross-attention outputs. Intra-layer edges capture token-level dependencies, while inter-layer edges track how representations flow between layers. To prevent misalignment accumulation, DAGR introduces three regularization losses: (i) Edge Consistency Loss which stabilizes local token interactions across tasks using KL divergence, (ii) Multi-hop Edge Consistency which extends stability constraints to multi-step reasoning chains, and (iii) Path Consistency Loss which enforces coherence in how information is transmitted across layers. The method is further enhanced with graph pruning and progressive regularization to improve efficiency on large-scale tasks. Experiments on the MTIL benchmark are provided to show that DAGR outperforms prior methods while using far fewer trainable parameters and no rehearsal data. Analytic experiments show that DAGR also is robust with respect to task order variations. Theoretical analysis is provided for stability guarantees by proving how graph regularization bounds representational drift across layers.

**Strengths:**

1. Treating modality alignment as a dynamic process rather than a static process and then modeling this dynamic process with graph-based regularization across layers and tasks is new and interesting.

2. Comparative experiments demonstrate consistent performance improvements compared to existing methods.

3. Full ablative experiments are offered to demonstrate that all aspects of the solution contribute to the optimal performance.

4. Analytic experiments on task order and training cots are informative and provide insight about DAGR.

5. The code is provided which makes reproducing the results straightforward.

**Weaknesses:**

1. The paper uses the idea that modality alignment is dynamic. But prior works such as Phang et al., 2021, Aitken et al., 2022, and Sun et al., 2025 have explored layer-wise feature drift and representational changes in multimodal and unimodal transformers. Hence, the core idea is not very novel.

2. Node construction in the DAG depends on clustering cross-modal embeddings from image and text branches. The number of clusters, fusion ratio can influence the graph structure and regularization effectiveness. Hence, users might need to carefully tune hyperparameters for each modality or dataset. Experiments only consider one dataset and it is unclear if it is easy to get good results easily.

3. Theoretical analysis on perturbation propagation seems mathematically sound, but similar perturbation sensitivity analyses have been done before for stability in deep networks and hence the results are not very novel and do not add much to the contributions of this work.

4. Although DAGR reduces forgetting through task-specific prompts, it still maintains one prompt per task, requiring storage proportional to the number of tasks. If we have many tasks, prompt storage and selection mechanisms need to be made memory-efficient and adaptive.

5. Experiments use CLIP which is a small model with today's standards. On the other hand, graph construction and regularization framework adds substantial computational overhead compared to simpler prompt-tuning methods. Hence, the approach may not scale efficiently to large transformers or high-dimensional tasks.

**Questions:**

1. Figure 5 shows that training takes hours. Wouldn't this make the proposed framework impractical for CL? Generally, it is necessary to adapt fast in CL.

2. Are there any evidence that the dynamic alignment challenge exists for different architectures or is it specific to CLIP-style multimodal architectures?

3. How scalable is DAGR’s graph construction and regularization to large number of tasks?

4. If one modality (e.g., text or image) is missing, corrupted, or partially misaligned, can DAGR still maintain stability? I have seen other papers on multimodal CL study this scenario because it can be common in practice.

---

### Official Review · Reviewer_6Ebc · 2025-11-01

**Soundness:** 3
**Presentation:** 3
**Contribution:** 3
**Rating:** 6
**Confidence:** 3

**Summary:**

This paper tackles the problem of multimodal continual learning, emphasizing that modality alignment is inherently dynamic and evolves with task learning and feature propagation across network layers. To address this, the authors propose the Dynamic Alignment Graph Regularization (DAGR) approach, a method that explicitly models evolving cross-modal alignments by introducing multi-level graph regularization across layers. Extensive experiments and ablation studies are conducted, demonstrating the effectiveness of the proposed methods and the contributions of each component.

**Strengths:**

+ The paper proposes the novel concepts of dynamic alignment in multimodal continual learning, which has not been considered in the literature before.
+ The proposed Dynamic Alignment Graph Regularization is technically sound and interesting, constructing graphs that capture both inter-layer and intra-layer feature relations. This approach effectively mitigates catastrophic forgetting by regularizing the consistency of these graphs across tasks.
+ The quantitative results are compelling, showing that the proposed method achieves superior performance and nearly reaches the upper bound on the MTIL benchmark.
+ The detailed analyses and ablation studies are conducted to validate the contribution of the designed components.
+ The proposed method is shown to be robust to the order of incremental tasks, a factor that often significantly affects performance in continual learning.

**Weaknesses:**

- Some important implementation details are missing, such as the backbone architecture used and the specifics of the Image–Language Matching (ILM) loss, which are essential for reproducibility and better understanding of the proposed framework.
- The generalizability of DAGR to different multimodal architectures remains unclear. The proposed method appears to be tailored for CLIP-like models with cross-attention modules, where graph regularization is applied to model cross-modal interactions. However, multimodal architectures can vary substantially, including single-stream multimodal transformers (e.g., ViLT) and multimodal large language models (MLLMs). It is therefore uncertain whether the proposed graph design can be effectively adapted or generalized to these alternative architectures.
- Scalability to more than two modalities (e.g., beyond two) is a potential concern. While the paper targets multimodal continual learning, it is limited to two modalities. The proposed graph regularization, being built on a pairwise cross-modal structure, may not readily extend to settings with three or more modalities.

**Questions:**

Please see the concerns in the weaknesses section.

---

### Official Review · Reviewer_8QAz · 2025-11-01

**Soundness:** 2
**Presentation:** 3
**Contribution:** 2
**Rating:** 4
**Confidence:** 4

**Summary:**

This paper introduces Dynamic Alignment Graph Regularization (DAGR), a novel method for Multimodal Continual Learning (MMCL) designed to mitigate catastrophic forgetting.
DAGR addresses modality alignment by constructing a graph on features and applies multi-level graph regularization to maintain consistency at the layer and task level.
The method is evaluated on the MTIL benchmark, but I have concerns with the experiments.

**Strengths:**

1. The proposed method of constructing features as graphs offers an interesting and potentially valuable perspective for multimodal continual learning.
2. The motivation for addressing dynamic modality alignment across sequential tasks is compelling and addresses a relevant challenge in multimodal continual learning.

**Weaknesses:**

1. The experimental evaluation has several critical limitations:
- (a) Missing state-of-the-art baselines: The majority of compared methods are unimodal continual learning approaches. The paper lacks comparisons with recent multimodal continual learning methods, particularly MoEAdapters4CL [1], which demonstrates superior performance on the same benchmarks (e.g., 54.3 vs. 53.5 on one metric, 87.4 vs. 87.1 on another). This absence raises concerns about whether the proposed method achieves state-of-the-art performance.

    [1] Yu, Jiazuo, et al. "Boosting continual learning of vision-language models via mixture-of-experts adapters." Proceedings of the IEEE/CVF Conference on Computer Vision and Pattern Recognition. 2024.

- (b) Incomplete task order analysis: The paper does not provide a comprehensive performance table across different task orders for MTIL Order II, as presented in ZSCL and other standard benchmarks. A thorough analysis of method performance under various task orderings is essential to demonstrate robustness, as performance may vary significantly with different task sequences.

2. A minor issue is that, the paper states that "existing methods often treat modality alignment as a static process" and that methods like weight regularization (Kirkpatrick et al., 2017) and incremental learning strategies (Rebuffi et al., 2017) "neglect dynamic adaptability challenges." However, the issue of representation change across tasks has been explicitly addressed in recent work, such as Ni et al. (ICML 2023) on "Continual vision-language representation learning with off-diagonal information".
Furthermore, the section titled MMCL but it mentions too many unimodal methods instead of multimodal ones. Please consider to discuss more mmcl methods as well.

3. Another concern is the paper fails to describe how key hyperparameters are tried and chosen, specifically the regularization weights and fusion ratio alpha. The performance of regularization-based methods is often highly sensitive to these weights.

**Questions:**

See Weaknesses

---

### Official Review · Reviewer_SjRG · 2025-11-11

**Soundness:** 2
**Presentation:** 3
**Contribution:** 2
**Rating:** 4
**Confidence:** 4

**Summary:**

This paper addresses the problem of catastrophic forgetting in multimodal continual learning (MMCL) by questioning the traditional view of static modality alignment. The work introduces Dynamic Alignment Graph Regularization (DAGR): a graph-based framework that models alignment across and within layers as a dynamic process, using multi-level graph regularization to mitigate shallow-layer misalignments propagating to deep layers. T

**Strengths:**

- The idea of ​​considering and placing constraints at the layer level is quite interesting.

- Intuitive illustrations and formulas help improve readability.

- Current experimental results on datasets show the superiority of the proposed method over several baselines.

**Weaknesses:**

1. It seems that the proposed method is like a type of distillation, going into each layer, and is quite expensive? Especially in an environment where data changes constantly, and the model needs to be updated quickly.

2. To date, there are many related works, which also propose multimodel for continual learning that are not mentioned and compared in the paper, especially the recent SOTAs [1, 2, 3, 4]

3. The paper would be more complete if the author considered comparing and presenting experiments on popular settings such as CIL (Class Incremental Learning), to show the correlation between the proposed method and recent SOTAs. Most of the datasets used seem to be more suitable for CIL (Class Incremental Learning) scenario than DIL (Domain Incremental Learning)

4. In lines 076-077, the authors claimed: "ensuring stable knowledge transfer when learning new tasks". It would be better if there were experimental results to illustrate this.

[1] "Mind the Gap: Preserving and Compensating for the Modality Gap in CLIP-Based Continual Learning", ICCV25.

[2] "External Knowledge Injection for CLIP-Based Class-Incremental Learning", ICCV25.

[3] "LADA: Scalable Label-Specific CLIP Adapter for Continual Learning", ICML25.

[4] "C-CLIP: Multimodal Continual Learning for Vision-Language Model", ICLR25.

**Questions:**

1. What is the relationship between the attention rollout matrix R and W(l) in Eq 7?

2. Do we need to build and save nodes layer by layer, for each task? Furthermore, does this method also need to save checkpoints of the old model? If so, logically, the computational cost seems to increase significantly compared to other methods. But it is not the case showed in Figure 5.

3. In addition to Figure 3, it would be more complete if the authors provided a visualization showing the changes through tasks, and compared to a specific method. What is the baseline that were mentioned? Is it the regular ViT with prompt or the CLIP model with prompt?

4. The authors should report additional forgetting metrics to demonstrate the effectiveness of the proposed method.

---

### Note · Authors · 2025-11-14

I have read and agree with the venue's withdrawal policy on behalf of myself and my co-authors.